# Proteomic Insights into Osteoporosis: Unraveling Diagnostic Markers of and Therapeutic Targets for the Metabolic Bone Disease

**DOI:** 10.3390/biom14050554

**Published:** 2024-05-04

**Authors:** Jihan Wang, Mengju Xue, Ya Hu, Jingwen Li, Zhenzhen Li, Yangyang Wang

**Affiliations:** 1Xi’an Key Laboratory of Stem Cell and Regenerative Medicine, Institute of Medical Research, Northwestern Polytechnical University, Xi’an 710072, China; jihanwang@nwpu.edu.cn (J.W.);; 2School of Medicine, Xi’an International University, Xi’an 710077, China; 3Department of Medical College, Hunan Polytechnic of Environment and Biology, Hengyang 421000, China; 4Research and Development Institute of Northwestern Polytechnical University in Shenzhen, Shenzhen 518057, China; 5School of Electronics and Information, Northwestern Polytechnical University, Xi’an 710129, China

**Keywords:** osteoporosis, bone metabolism, proteomics, diagnostic biomarkers, therapeutic targets

## Abstract

Osteoporosis (OP), a prevalent skeletal disorder characterized by compromised bone strength and increased susceptibility to fractures, poses a significant public health concern. This review aims to provide a comprehensive analysis of the current state of research in the field, focusing on the application of proteomic techniques to elucidate diagnostic markers and therapeutic targets for OP. The integration of cutting-edge proteomic technologies has enabled the identification and quantification of proteins associated with bone metabolism, leading to a deeper understanding of the molecular mechanisms underlying OP. In this review, we systematically examine recent advancements in proteomic studies related to OP, emphasizing the identification of potential biomarkers for OP diagnosis and the discovery of novel therapeutic targets. Additionally, we discuss the challenges and future directions in the field, highlighting the potential impact of proteomic research in transforming the landscape of OP diagnosis and treatment.

## 1. Introduction

Osteoporosis (OP), a prevalent and debilitating skeletal disorder, poses a significant public health concern globally, particularly among aging populations [1,2,3]. Characterized by compromised bone strength and increased fracture risk, OP not only diminishes individuals’ quality of life but also imposes substantial economic burdens on healthcare systems. The traditional diagnostic methods and treatment strategies for OP have long served as the cornerstone of clinical management. Conventional diagnostic approaches primarily rely on bone mineral density (BMD) measurements obtained through dual-energy X-ray absorptiometry (DXA), supplemented by clinical risk assessment tools such as FRAX [4,5,6,7]. While DXA remains the gold standard for OP diagnosis, its limitations in assessing bone quality and fracture risk highlight the need for complementary approaches. Likewise, current treatment modalities predominantly focus on anti-resorptive agents, such as bisphosphonates, and anabolic agents, like teriparatide, aimed at mitigating bone loss and enhancing bone formation. However, despite their efficacy in reducing fracture risk, these treatments are not devoid of limitations, including potential adverse effects and suboptimal response rates. Thus, there exists a pressing demand for novel diagnostic methodologies and therapeutic interventions that can provide more accurate risk stratification, personalized treatment strategies, and improved patient outcomes.

The application of proteomic approaches in OP research presents a compelling rationale rooted in the complex and dynamic nature of bone biology [8,9]. Traditional methods often fail to capture the intricate molecular mechanisms underlying bone metabolism and homeostasis, limiting our understanding of disease pathogenesis and hindering the development of effective diagnostic markers and therapeutic targets. Proteomics, with its ability to comprehensively analyze the entire repertoire of proteins expressed in a biological system, offers unprecedented insights into the intricate protein networks governing bone health [10]. By elucidating the proteomic landscape of OP, researchers can identify potential biomarkers for early disease detection, unravel novel pathways implicated in bone remodeling, and pinpoint therapeutic targets for precision medicine interventions. Furthermore, proteomic technologies enable the exploration of post-translational modifications (PTMs) and protein–protein interactions, providing a deeper understanding of the molecular events driving disease [11,12,13,14]. This review will delineate the rationale for leveraging proteomic approaches in OP research, highlighting their potential to revolutionize diagnosis, prognosis, and treatment strategies for this debilitating disease.

## 2. Proteomic Technologies in Osteoporosis Research

In recent years, proteomic technologies have emerged as powerful tools for unraveling the complexities of OP. We delve into a diverse array of proteomic methodologies, including mass spectrometry (MS)-based approaches, gel-based and gel-free techniques, and the integration of bioinformatics tools for data analysis and interpretation, as illustrated in Figure 1. By elucidating the bone proteome and uncovering molecular mechanisms underlying OP, proteomic studies hold great promise for revolutionizing the diagnosis, treatment, and management of this prevalent skeletal disease.

### 2.1. Mass Spectrometry-Based Proteomics

Mass spectrometry (MS) has emerged as a powerful tool in OP research, enabling comprehensive analysis of the bone proteome. MS-based proteomics facilitates the identification and quantification of proteins involved in bone metabolism, providing valuable insights into disease mechanisms and potential therapeutic targets. For instance, liquid chromatography–tandem mass spectrometry (LC-MS/MS) allows for the identification of hundreds of proteins in bone tissues and biofluids, uncovering novel biomarkers associated with OP pathogenesis [15,16,17]. Moreover, targeted MS techniques, such as selected reaction monitoring (SRM) and parallel reaction monitoring (PRM), enable precise quantification of specific proteins of interest, offering opportunities for biomarker validation and personalized medicine approaches in OP research [18,19,20]. MS-based proteomics also facilitates the characterization of post-translational modifications (PTMs) in bone proteins, providing insights into their functional roles and regulatory mechanisms in OP [21,22,23]. Overall, MS-based proteomics hold great promise for advancing our understanding of osteoporosis and identifying diagnostic markers and therapeutic targets for the disease.

### 2.2. Gel-Based and Gel-Free Proteomic Techniques

Gel-based and gel-free proteomic techniques complement MS-based approaches by enabling protein separation, enrichment, and analysis [24,25,26,27]. Bone metabolism is balanced by bone-forming osteoblasts and bone-resorbing osteoclasts. Gel-based and gel-free proteomic approaches help enhance our comprehension of post-genomic mechanisms underlying bone metabolic dysfunctions in OP [28,29,30,31]. Researchers using two-dimensional gel electrophoresis (2-DE) and isotope-coded affinity tags (ICAT) analysis identified 23 spots in 2-DE and 19 proteins in ICAT, which were expressed differently during osteoclast differentiation. These two methods gave us closely related but different information about proteins, suggesting they are complementary or at least supplementary methods [32]. Additionally, gel-free techniques such as shotgun proteomics and label-free quantification methods offer high-throughput analysis of complex protein mixtures, expanding the coverage of the bone proteome [33,34,35]. These techniques enable the discovery of novel biomarkers and signaling pathways associated with OP pathogenesis, enhancing our understanding of disease mechanisms. Furthermore, gel-based and gel-free proteomic approaches provide complementary information on protein abundance, PTMs, and protein–protein interactions, contributing to a comprehensive characterization of the bone proteome in health and disease.

### 2.3. Bioinformatics Tools for Proteomic Data Analysis and Interpretation

The vast amount of data generated from proteomic experiments requires sophisticated bioinformatics tools for data processing, analysis, and interpretation. Bioinformatics platforms offer solutions for protein identification, quantification, functional annotation, and pathway analysis, facilitating the extraction of meaningful insights from complex proteomic datasets. Database search algorithms, including Mascot [36,37], SEQUEST [38,39], and MaxQuant [40,41], are commonly used for peptide identification, while label-free or stable isotope labeling-based quantification methods rely on bioinformatics tools for accurate protein quantification. Moreover, bioinformatics tools enable the functional annotation and pathway analysis of identified proteins, shedding light on the biological processes perturbed in OP. Gene Ontology (GO) [42,43] analysis categorizes proteins based on their biological process (BP), molecular function (MF), and cellular component (CC), while pathway analysis tools such as Kyoto Encyclopedia of Genes and Genomes (KEGG) [44,45] and Reactome [46,47] elucidate the interconnected signaling pathways dysregulated in OP. Ingenuity Pathway Analysis (IPA) [48,49], STRING [50,51], and the Database for Annotation, Visualization and Integrated Discovery (DAVID) [52] provide functional annotation and enrichment analysis, revealing biological pathways and protein–protein interaction (PPI) networks associated with OP pathogenesis. Integrating bioinformatics tools with proteomic technologies enhances the discovery of diagnostic markers and therapeutic targets for osteoporosis, facilitating translational research efforts aimed at improving patient care.

## 3. Proteomic Insights into Bone Metabolism

### 3.1. Identification of Key Proteins Involved in Bone Formation and Resorption

Proteomic approaches have greatly facilitated the identification and characterization of key proteins involved in bone formation and resorption, which are critical processes in maintaining bone homeostasis. Various studies have utilized MS-based proteomic techniques to analyze bone tissues, cell cultures, and biofluids to identify proteins associated with osteoblastogenesis, osteoclastogenesis, and bone remodeling pathways. For instance, in a study exploring the proteins bound to the osteocalcin OSE2 sequence of the mouse osteocalcin promoter, TRPS1 was identified as a crucial regulator of osteocalcin transcription [53]. Human mesenchymal stem cells from bone marrow (BM-hMSCs) are widely recognized as ideal candidates for bone tissue engineering. Proteomic analysis revealed that extracellular calcium ions (Ca^2+^) notably enhanced the proliferation of BM-hMSCs. Furthermore, it suggested that the MAPK signaling pathway might be involved in the Ca^2+^-induced osteogenic differentiation of BM-hMSCs [54]. Estrogen is known to play a significant role in inhibiting osteoclast differentiation and protecting against osteoporosis-related bone loss, particularly in postmenopausal women. A proteomic analysis elucidated estrogen’s inhibitory effect on osteoclasts, highlighting its role in promoting apoptosis, suppressing differentiation, and preventing the polarization of osteoclasts [55]. The proteomic profile of osteoclast membrane proteins identified Nhedc2 as a key player in proton transport within osteoclasts, thereby regulating their function [56]. Similarly, a proteomic analysis of serum-derived exosomes from individuals with neurofibromatosis type 1 congenital tibial pseudarthrosis demonstrated their detrimental impact on bone by promoting osteoclastogenesis and inhibiting osteoblastogenesis [57]. Additionally, a proteomic investigation identified LBP and CD14 as pivotal proteins involved in interactions between blood and biphasic calcium phosphate microparticles [58]. These proteomic insights provide valuable knowledge of the molecular mechanisms underlying bone metabolism, offering potential targets for therapeutic interventions in OP [59,60,61,62].

### 3.2. Quantitative Proteomics in Assessing Dynamic Changes in Bone Proteome

Quantitative proteomic techniques have emerged as powerful tools for assessing dynamic changes in the bone proteome under physiological and pathological conditions. Stable isotope labeling-based methods, such as tandem mass tag (TMT) [63,64] and isobaric tags for the relative and absolute quantitation (iTRAQ) [65], allow for the simultaneous quantification of thousands of proteins across multiple samples, enabling the comprehensive analysis of bone proteome alterations [66,67,68]. Several quantitative proteomic studies have investigated changes in protein expression profiles during bone development, aging, and osteoporosis progression [18,59,69]. Recently, a novel mass spectrometric workflow has been introduced to explore the bone proteome, offering deep coverage and quantification strategies. This approach unveils key components like extracellular matrix proteins, bone-specific proteins, and signaling molecules while also identifying post-translational modifications and senescence factors relevant to age-related bone diseases [70]. Moreover, quantitative proteomic analyses have been instrumental in elucidating the effects of therapeutic interventions, such as bisphosphonate treatment, on the bone proteome, providing insights into their mechanisms of action and potential side effects [71].

### 3.3. Proteomic Studies on the Bone Extracellular Matrix

Extracellular matrix (ECM) dynamics represent an emerging yet understudied hallmark of aging and longevity [72]. The bone ECM encompasses mineral deposits on extensively crosslinked collagen fibrils alongside numerous non-collagenous proteins. Many of these proteins are pivotal in governing bone formation and regeneration through signaling pathways, contributing crucial regulatory and structural functions [73]. Proteomic studies focusing on the bone ECM have shed light on the composition, organization, and remodeling dynamics of this essential structural component. The bone ECM provides a dynamic scaffold for bone cells and plays a crucial role in regulating bone architecture, mineralization, and mechanical properties. Proteomic analyses of the bone ECM have identified numerous ECM-associated proteins, including collagens [74,75], proteoglycans [76,77], glycoproteins [76,78], and matricellular proteins [79], which contribute to ECM integrity and function. For instance, studies have identified collagen isoforms and post-translational modifications that influence bone matrix assembly and mineralization [80,81]. Additionally, proteomic analyses have revealed ECM proteins involved in cell–ECM interactions and signaling pathways, such as integrins and focal adhesion kinases, which play pivotal roles in regulating osteoblast and osteoclast behavior [82,83,84,85,86]. Understanding the composition and regulation of the bone ECM through proteomic approaches provides valuable insights into the pathogenesis of osteoporosis and offers potential targets for ECM-targeted therapies to enhance bone strength and integrity.

### 3.4. Advancements in Proteomics Related to Rare Bone Diseases

In research on musculoskeletal health in children and adolescents, proteomic and transcriptomic studies have identified the skeletal muscle “secretoma,” consisting of several myokines with endocrine and paracrine functions [87]. Additionally, these studies offer a comprehensive overview of the pathobiology of fibrodysplasia ossificans progressiva (FOP), highlighting advances in molecular genetics and proteomic research [88,89]. Furthermore, proteomic analysis has been instrumental in identifying predictive plasma biomarkers, such as ceruloplasmin and immunoglobulins, for metastatic Ewing’s sarcoma in children [90]. Moreover, label-free quantitative proteomics analysis of fluconazole-treated Dental Pulp Mesenchymal Stem/stromal cells from CA-II-deficient osteopetrosis patients has revealed potential treatment avenues for osteopetrosis by identifying differentially expressed proteins, including ATP1A2, CPOX, Ap2 alpha, RAP1B, and members of the RAB protein family [91]. These above findings underscore the pivotal role of proteomic analysis in elucidating molecular mechanisms and identifying therapeutic targets for rare bone diseases.

## 4. Diagnostic Markers in Osteoporosis

### 4.1. Blood and Urinary Biomarkers

Blood and urinary biomarkers are pivotal in diagnosing osteoporosis, offering valuable insights into bone turnover and fracture risk prediction. These biomarkers encompass molecules associated with both bone formation, such as osteocalcin (OC) and bone alkaline phosphatase (BALP), and bone resorption, including C-terminal telopeptide of type I collagen (CTx) and N-terminal telopeptide of type I collagen (NTx). Elevated levels of these biomarkers in serum and urine have been consistently correlated with heightened fracture risk and bone loss in individuals with osteoporosis [92,93,94]. Additionally, assessing the ratio of bone resorption to formation markers, such as the osteoprotegerin/receptor activator of nuclear factor kappa-B ligand (OPG/RANKL) ratio, provides further valuable information regarding bone metabolism status and fracture risk [95,96]. This comprehensive analysis of blood and urinary biomarkers serves as a critical component in the multifaceted approach to diagnosing and managing OP, enabling clinicians to better understand disease progression and tailor treatment strategies effectively. Furthermore, this compilation holds significant importance as it consolidates the advancements in proteomic technology applied to OP biomarker research. Proteomic approaches offer the potential to identify novel diagnostic markers and proteomic signatures for the early detection of OP. By analyzing the protein expression profiles in serum, urine, and bone tissues, proteomic studies have identified candidate biomarkers associated with OP pathogenesis and fracture risk. Integrating proteomic signatures with traditional diagnostic markers may improve the accuracy and sensitivity of OP diagnosis, enabling early intervention and prevention strategies. By summarizing recent findings in Table 1, researchers and clinicians gain valuable insights into the evolving landscape of diagnostic and prognostic markers for OP. This synthesis facilitates a deeper understanding of the molecular mechanisms underlying bone turnover and fracture risk, paving the way for more precise diagnostic approaches and targeted therapeutic interventions. Additionally, it underscores the potential for proteomic technology to unravel novel biomarkers that may enhance early detection, risk stratification, and personalized management strategies for individuals at risk of OP-related complications.

### 4.2. Potential Integration with Imaging Techniques for Comprehensive Diagnosis

The integration of proteomic biomarkers with imaging techniques holds promise for comprehensive diagnosis and risk assessment of OP. DXA remains the gold standard for assessing BMD and diagnosing OP; however, it has limitations in predicting fracture risk and assessing bone quality. Complementary imaging techniques such as quantitative computed tomography (QCT) [122,123,124,125], high-resolution peripheral quantitative computed tomography (HR-pQCT) [126,127,128], and magnetic resonance imaging (MRI) [129,130,131] provide additional information on bone microarchitecture, volumetric BMD, and bone strength. By integrating proteomic biomarkers with advanced imaging modalities, clinicians can obtain a comprehensive evaluation of bone health status, including both structural and molecular aspects. For instance, the multicenter prospective cohort study developed a comprehensive fracture risk assessment tool incorporating proteomic, genomic, and clinical factors with the objective of enhancing fracture prediction accuracy across diverse populations. By integrating proteomic data from blood samples with imaging techniques, such as HR-pQCT, the study seeks to improve the diagnosis and management of OP and OF, ultimately advancing personalized medicine approaches for fracture prevention [132]. This multidimensional approach may enhance the accuracy of OP diagnosis, risk stratification, and treatment monitoring, ultimately improving patient outcomes.

## 5. Therapeutic Targets and Drug Discovery

### 5.1. Proteomic Identification of Novel Therapeutic Targets

Proteomic approaches play a pivotal role in identifying novel therapeutic targets for OP by unraveling the intricate molecular mechanisms underlying bone metabolism [15,133,134]. By analyzing the proteomic landscape of bone tissues, cell cultures, and biofluids, researchers can identify dysregulated proteins and pathways associated with OP pathogenesis. For instance, a study has identified seven circulating proteins, including ANTXR2, cadherin-13 (CDH-13), CD163, COMP, DKK3, periostin, and secretogranin-1, which decrease with age in mice. Among these, CDH-13 was found to inhibit osteoclast differentiation and delay age-related bone loss in aged mice, suggesting its potential therapeutic role in preventing osteopenia [120]. Another study illustrated, through proteomic analysis, that estrogen promotes autophagy in human osteoblasts during differentiation to promote survival and mineralization by upregulating RAB3 GTPase-activating protein [135,136]. Furthermore, through proteomic techniques and functional validation, ubiquitin C-terminal hydrolase 1 (UCHL1) was shown to stabilize the transcriptional coactivator with PDZ-binding motif (TAZ) by deubiquitination, inhibiting osteoclast formation. These findings collectively highlight UCHL1 as a potential therapeutic target for osteoporosis by modulating bone remodeling processes [137].

Traditional Chinese medicine (TCM) has demonstrated efficacy in treating human diseases over more than two millennia, underscoring its enduring value in healthcare. Proteomic studies serve as powerful tools in elucidating the mechanistic basis of TCM therapies, offering valuable insights into the molecular pathways underlying their therapeutic effects. In an osteoporosis study, researchers employed proteomic analysis, specifically iTRAQ technology, to identify potential therapeutic targets associated with deer antler extract. Through comprehensive serum protein profiling and bioinformatics analysis, they revealed a complex interaction network involving various proteins and signaling pathways related to bone formation and remodeling [138]. Another investigation utilized proteomic analysis to explore the effects of Er-Xian Decoction (EXD) on osteoblastic and osteoclastic cells. Their findings demonstrated a significant modulation of protein expressions associated with osteoblastic proliferation, differentiation, and apoptosis, as well as osteoclastic protein folding and aggregation [139]. Metabolomics combined with proteomics analysis of the femur provided a comprehensive interpretation of the changes in PMOP under salidroside treatment [140]. Furthermore, a study investigating the mechanism of action of Bugu Shengsui Decoction in treating osteoporosis revealed that the decoction promotes osteogenesis via the PI3K-AKT pathway, highlighting its potential as a therapeutic target for osteoporosis [141]. By targeting these dysregulated proteins and pathways, novel therapeutic interventions can be developed to restore bone homeostasis and prevent fractures in osteoporotic patients.

### 5.2. Evaluation of Proteomic Profiles in Drug Development and Personalized Medicine

Proteomic data serve as valuable resources for evaluating the efficacy and safety of potential drug candidates in drug development [142,143]. Proteomic data are uniquely valuable for predicting drug responses and discovering biomarkers since drugs primarily interact with proteins in target cells rather than with DNA or RNA [144]. By integrating proteomic analyses with preclinical and clinical studies, researchers can assess the effects of candidate drugs on the bone proteome and identify molecular mechanisms underlying their therapeutic actions. For example, studies have identified serum proteome markers associated with the response to antiosteoporosis drugs, teriparatide and denosumab, in osteoporosis patients. Proteomic analysis revealed significant changes in protein levels post-treatment, particularly in pathways related to insulin-like growth factor 1 (IGF-I) and the innate immune system, suggesting their involvement in drug response [145]. Another study comprehensively investigated the anti-osteoporotic mechanism of icariin by integrating proteomics and NMR metabonomics, and the results revealed significant alterations in proteins and metabolites related to various pathways, including bone remodeling, energy metabolism, and signaling pathways [146]. Moreover, proteomic analyses can identify biomarkers of drug response and adverse effects, enabling personalized treatment strategies and improving patient outcomes [147,148,149]. Integrating proteomic data into the drug development process enhances the understanding of drug mechanisms, accelerates the identification of promising candidates, and minimizes the risk of treatment-related complications. Furthermore, genomic or proteomic profiling can identify patients at high risk of treatment failure or adverse drug reactions, facilitating early intervention and personalized monitoring strategies [133,150,151]. Overall, personalized medicine approaches based on proteomic insights have the potential to revolutionize osteoporosis management by improving treatment efficacy, minimizing side effects, and reducing fracture risk in vulnerable patient populations.

## 6. Challenges and Future Directions

### 6.1. Limitations and Challenges in Proteomic Studies of Osteoporosis

Despite the promise of proteomic technologies in unraveling the complexities of osteoporosis, several limitations and challenges hinder their application and interpretation. One significant challenge is the heterogeneity of bone tissues and the dynamic nature of bone metabolism, which can introduce variability in proteomic analyses. Sample collection and processing methods, including sample storage conditions and protein extraction protocols, may also impact the reproducibility and reliability of proteomic data [152,153]. Moreover, the complexity of the bone proteome, including the presence of low-abundance proteins and PTMs, poses analytical challenges for protein identification and quantification [154,155]. Additionally, the standardization of proteomic workflows, data analysis pipelines, and quality control (QC) measures is essential to ensure the robustness and reproducibility of proteomic studies in osteoporosis research [156,157].

### 6.2. Integration of Multi-Omic Data for a Holistic Understanding

To overcome the limitations of proteomic studies and achieve a comprehensive understanding of osteoporosis, the integration of multi-omic data is imperative. Combining proteomic data with genomic, transcriptomic, metabolomic, and epigenomic datasets can provide a holistic view of the molecular mechanisms underlying osteoporosis pathogenesis and treatment response. The integration of multi-omics data enables the identification of key regulatory pathways, biomolecular networks, and potential therapeutic targets that may not be evident from individual omics analyses alone [158,159,160]. For example, integrating proteomic and transcriptomic data can elucidate the relationship between gene expression and protein abundance, revealing post-transcriptional regulatory mechanisms in osteoporosis [133,161,162]. Similarly, the integration of proteomic and metabolomic data can uncover metabolic pathways dysregulated in osteoporosis and identify metabolic biomarkers associated with disease progression and fracture risk [146]. Overall, the integration of multi-omics data holds promise for advancing our understanding of osteoporosis pathophysiology and facilitating the development of personalized therapeutic strategies [163,164].

### 6.3. Future Directions and Potential Impact on Clinical Practice

Looking ahead, several future directions in proteomic research have the potential to transform clinical practice in osteoporosis management. Firstly, advancements in proteomic technologies, such as improved instrumentation, sample preparation methods, and data analysis algorithms, will enhance the sensitivity, specificity, and throughput of proteomic studies in osteoporosis. Secondly, the development of targeted proteomic assays, including MRM and PRM, will enable the accurate quantification of specific protein biomarkers in clinical samples, facilitating their translation into diagnostic tests and therapeutic monitoring tools [165,166,167]. Additionally, the integration of proteomic biomarkers into clinical decision-making algorithms and risk assessment models may improve the accuracy of osteoporosis diagnosis, fracture prediction, and treatment selection. Furthermore, we recognize that various factors, including study design, sample size, patient demographics, and methodological variances, play crucial roles in influencing the identification and validation of biomarkers in proteomic screening approaches. These elements contribute to the observed discrepancies in biomarker identification across different studies, as exemplified by the case of CDH-13 in the above reference. It is imperative to critically evaluate these factors to discern the clinical significance of putative biomarkers in OP. Finally, the implementation of personalized medicine approaches based on proteomic profiling may enable tailored interventions targeting individual patient needs and disease characteristics, ultimately improving patient outcomes and reducing the burden of osteoporosis-related fractures [168,169]. We emphasize the necessity for additional research endeavors aimed at validating and replicating findings across independent cohorts. Robust study designs and methodologies are essential for elucidating the true clinical relevance of these biomarkers in the intricate pathogenesis and management of OP. By addressing these considerations, we aim to provide a more comprehensive understanding of the potential implications of our findings in the field of OP research.

## 7. Conclusions

In summary, proteomic research has emerged as a powerful tool for unraveling the complex molecular mechanisms underlying osteoporosis, offering valuable insights into disease pathogenesis and potential therapeutic targets (Figure 1). Through proteomic analyses of bone tissues, biofluids, and circulating cells, researchers have identified novel diagnostic markers, therapeutic targets, and molecular pathways implicated in osteoporosis development and progression. The identification of key proteins involved in bone metabolism, such as regulators of osteoblast and osteoclast activity, has provided a deeper understanding of bone homeostasis and remodeling processes. Moreover, proteomic studies have facilitated the discovery of potential biomarkers for the early detection of osteoporosis and prediction of fracture risk, enabling timely intervention and personalized treatment strategies. Despite the challenges and limitations inherent in proteomic studies, including sample heterogeneity and analytical complexity, ongoing advancements in proteomic technologies and data integration approaches hold promise for furthering our understanding of osteoporosis and translating proteomic insights into clinical practice.

The transformative potential of proteomic research in osteoporosis extends beyond diagnosis to encompass therapeutic discovery and personalized medicine approaches. By identifying dysregulated proteins and pathways associated with osteoporosis pathogenesis, proteomic studies have unveiled promising therapeutic targets for drug development and intervention. The integration of multi-omics data, including genomics, transcriptomics, and metabolomics, offers a holistic understanding of osteoporosis at the molecular level, facilitating the development of targeted therapies and personalized treatment regimens. The translation of proteomic biomarkers into clinical practice has the potential to revolutionize osteoporosis management by improving diagnostic accuracy, guiding treatment selection, and monitoring treatment response. Ultimately, proteomic insights into osteoporosis promise to advance patient care, reduce the burden of osteoporotic fractures, and enhance the quality of life for individuals affected by this debilitating disease.

## Figures and Tables

**Figure 1 biomolecules-14-00554-f001:**
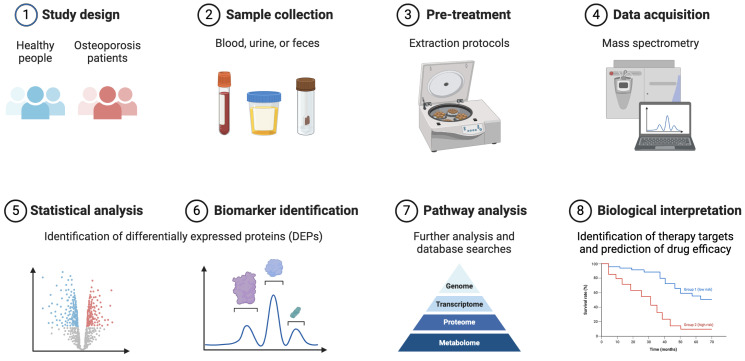
Summary view of proteomic analysis and application in osteoporosis. Proteomic analysis involves the comprehensive study of proteins expressed by cells, tissues, or biofluids, providing insights into their structure, function, and interactions. In the context of osteoporosis, proteomic techniques like mass spectrometry and protein microarrays systematically identify and characterize proteins associated with bone metabolism and homeostasis. By comparing protein expression patterns between healthy and osteoporosis patients, researchers can pinpoint biomarkers indicative of disease progression or treatment response.

**Table 1 biomolecules-14-00554-t001:** Summary of proteomic studies identifying biomarkers for osteoporosis.

Biomarkers	Sample Type	Proteomic Technology	Summary	Refs
MYH14, IGLC1, MEX3B, and FBLN1	Serum	LC-MS	This cohort study identified potential protein biomarkers associated with osteopenia (ON) and OP using LC-MS proteomics. Notably, MYH14, IGLC1, MEX3B, and FBLN1 were highlighted as key markers showing dysregulation in low BMD progression, with a focus on inflammatory pathways such as TNF, TLR4, and IFNG.	[97]
Lysozyme C, Glucosidase, Disulfide Isomerase A5	Plasma	LC–MS/MS, PRM	The expression of protein Lysozyme C was negatively related to BMD, while the expression of Glucosidase and Disulfide Isomerase A5 was positively related to BMD values.	[19]
Sox2, Oct3/4, Nanog, and E-cadherin	Blood-derived stem cells (BDSCs)	Proteome Profiler Array	Embryonic markers Sox2, Oct3/4, Nanog, and E-cadherin, which showed decreased expression during osteoblastic differentiation induced by rapamycin under microgravity conditions.	[98]
ABI1	Peripheral blood monocytes (PBMs), plasma	LC-MS/MS, Western Blotting (WB), ELISA	ABI1 was significantly down-regulated in PBM in Chinese elderly men with extremely low vs. high BMD, as well as in osteoporotic fracture (OF) patients vs. non-fractured (NF) subjects; the plasma ABI1 protein has superior performance in discriminating osteopenia and healthy subjects.	[99]
17 proteins	Serum exosomes	LC–MS	A total of 188, 224, and 185 proteins were identified in the normal, ON, and OP groups, respectively. There were 17 proteins significantly dysregulated in the ON and OP groups.	[34]
CHD1, PNP	Serum	4-D label-free proteomics, ELISA	Serum-level CHD1 and PNP have the potential power as effective indicators for the diagnosis of postmenopausal osteoporosis (PMOP)	[100]
Ubiquitylomes	Whole blood	high-performance liquid chromatography (HPLC), LC-MS/MS	This study identified differential ubiquitination patterns in whole blood between healthy postmenopausal women and PMOP patients, revealing potential biomarkers associated with PMOP. Key findings include dysregulation in ubiquitin-conjugating enzyme activity, enrichment in pathways such as ubiquitin-mediated proteolysis, and identification of potential diagnostic targets in whole blood.	[101]
An 18-peptide multidimensional OP urinary proteomic profile biomarker	Urine	capillary electrophoresis coupled with MS (CE-MS)	This study developed and validated an 18-peptide multidimensional urinary proteomic profile (OSTEO18) biomarker for osteoporosis in heart transplant recipients, showing promising diagnostic performance with improved accuracy compared to known risk factors.	[102]
VDBP	Serum	2-D gel electrophoresis, ELISA	This study identified 27 spots of interest when comparing low BMD versus normal BMD postmenopausal women, and low serum vitamin D-binding protein (VDBP) levels correlate with low BMD.	[103]
PSMB9, AARS, PCBP2, and VSIR	Plasma exosome	LC-nano-MS/MS, PRM	This study identified 45 differentially expressed proteins, and 4 of them (PSMB9, AARS, PCBP2, and VSIR) associated with osteoporosis were further verified.	[18]
Fibrinogen, vitronectin, clusterin, coagulation factors, and apolipoprotein	Extracellular vesicles (EVs)	nano-HPLC-ESI-MS/MS	The proteomic comparison between osteopenic and healthy controls EVs evidenced a decrease in fibrinogen, vitronectin, and clusterin and an increase in coagulation factors and apolipoprotein, which was also upregulated in OP EVs.	[104]
IL-6, LT-α, FLT3LG, CSF1, and CCL7	Serum	Target 48 Cytokine Panel	This observational study identified several serum cytokines, including Interleukin 6 (IL-6), Lymphotoxin-alpha (LT-α), Fms-related tyrosine kinase 3 ligand (FLT3LG), Colony stimulating factor 1 (CSF1), and Chemokine (C-C motif) ligand 7 (CCL7), as potential markers associated with hip fracture status in older adults.	[105]
PKM2	PBMs	LC-MS/MS	This study discovered 59 DEPs and validated the significant upregulation of pyruvate kinase isozyme 2 (PKM2) with OP.	[106]
ITIH4	Serum	Protein chip SELDI TOF-MS	This study identified specific serum protein peaks, notably fragments of interalpha-trypsin-inhibitor heavy chain H4 precursor (ITIH4), as potential biomarkers for discriminating between postmenopausal women with high or low/normal bone turnover.	[107]
AMFR	Plasma	Protein microarray, WB	Decreased levels of autocrine motility factor receptor (AMFR) were identified and validated in the blood plasma of female osteoporosis patients.	[108]
SOD, A1AT	Urine (Rats)	2-D gel, MS spectrometry	This study identified superoxide dismutase (SOD) as a down-regulated protein and alpha-1-antitrypsin (A1AT) as an upregulated protein in the urine of ovariectomized rats.	[109]
20 proteins	Serum	LC-IMS-MS	This study identified 20 proteins associated with accelerated BMD loss in older men, with five proteins also linked to incident hip fracture. Notable proteins included CD14, SHBG, B2MG, TIMP1, CO7, CO9, and CFAD, suggesting their potential as biomarkers for future research in bone biology and fracture prediction.	[110]
Four proteins	Serum	MALDI-TOF MS combined with WCX magnetic beads	This study identified four potential serum protein biomarkers for PMOP, including *m*/*z* peaks at 3167.4, 4071.1, 7771.7, and 8140.5, using MALDI-TOF MS combined with weak cationic exchange (WCX) magnetic beads.	[111]
NPM1, APMAP, COX6A1, and ACP5	Femur (Rats)	LC-MS/MS	A total of 47 differentially expressed proteins (DEPs) were identified in glucocortocoid-induced osteoporosis (GIOP) rats. Protein NPM1, APMAP, COX6A1, and ACP5 showed a close relationship with pathogenesis of GIOP, which could serve as potential biomarkers of GIOP.	[112]
CSC1-like protein, PTPN11, SLC44A1, and MME	Human bone marrow stromal cells (BMSCs)	LFQ nLC-MS/MS	This study identified dysregulated proteins, including CSC1-like protein, PTPN11, SLC44A1, and MME, in human bone marrow stromal cells exposed to simulated microgravity.	[113]
12 candidate biomarkers	Serum	Label-free LC-MS/MS	A panel of 12 candidate biomarkers was selected, of which 1 DEP (RYR1) was found upregulated in the osteopenia and OP groups, 8 DEPs (APOA1, SHBG, FETB, MASP1, PTK2B, KNG1, GSN, and B2M) were upregulated in OP and 3 DEPs (APOA2, RYR3, and HBD) were down-regulated in osteopenia or OP.	[114]
IL-7, CXCL-12, CXCL-8	Serum	Olink^®^ Target 48 Cytokine Panel	This study identified IL-7 and CXCL-12 as biomarkers associated with better functional recovery at three months after discharge, while CXCL-8 was associated with an increased risk of readmission in older adults with hip fractures.	[115]
HSP27	PBMs	4-plex iTRAQ coupled with LC-MS/MS	Levels of heat shock protein 27 (HSP27) were elevated in low-BMD conditions in both premenopausal and postmenopausal women.	[23]
Two proteins	Serum	MALDI-TOF-MS	This study identified two potential serum protein markers, with mass-to-charge ratios of 1699 Da and 3038 Da, for screening osteopenia in postmenopausal women.	[116]
HNP-1	Salivary fluid	MALDI TOF MS	Higher concentrations of α-defensin human neutrophil peptide-1 (HNP-1, a peptide released by neutrophils) were associated with lower BMD in postmenopausal women.	[117]
Four proteins	Plasma (Rats)	ESI-Q-TOF-MS, ESI-QqLIT-MS	This study identified four plasma proteins, including mannose-binding lectin-C, major urinary protein 2, type I collagen alpha 2 chain, and tetranectin, as significantly elevated in ovariectomized mice (ovx) compared to sham mice. Among these proteins, tetranectin showed a marked upregulation of almost 50 times in the ovx mice.	[20]
GHR, IGFBP2, GDF15, EGFR, CD14, CXCL12, MMP12, and ITIH3	Plasma	5 K SomaScan version 4.0 aptamer-based assay	This study identified several circulating proteins associated with incident hip fractures, including proteins related to the growth hormone/insulin growth factor system (GHR and IGFBP2), as well as GDF15, EGFR, CD14, CXCL12, MMP12, and ITIH3.	[118]
PINP	Plasma or serum (Rats)	LC-MS/MS	Circulating PINP levels in rats showed age-dependent changes, decreased with prednisolone treatment, and increased with parathyroid hormone (PTH) treatment, suggesting its potential as a biomarker for bone physiology in rat models of osteoporosis.	[119]
22 proteins	Serum	LC–MS/MS	22 proteins, including PHLD, SAMP, PEDF, HPTR, APOA1, SHBG, CO6, A2MG, CBPN, RAIN APOD, and THBG, were found to significantly correlate with BMD in OP.	[15]
CDH-13	Plasma (Mice)	MS	This study identified seven circulating proteins, including ANTXR2, CDH-13, CD163, COMP, DKK3, periostin, and secretogranin-1, which decrease with age in mice. Among these, CDH-13 was found to inhibit osteoclast differentiation and delay age-related bone loss in aged mice.	[120]
Proteomic profiling of human bone from different anatomical sites	Bone	LC-MS/MS	Results from this study revealed distinct protein profiles between alveolar bone (AB), iliac cortical (IC) bone, and iliac spongiosa (IS). AB exhibited an ECM protein-related fingerprint, while IS and IC displayed an immune-related proteome fingerprint.	[121]

Note: in Table 1, those not labeled “Rats” or “Mice” for the sample type are all human specimens.

## Data Availability

Not applicable.

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
