# Peer review of "Proteomic Insights into Osteoporosis: Unraveling Diagnostic Markers of and Therapeutic Targets for the Metabolic Bone Disease"

_biomolecules, 2024, doi:10.3390/biom14050554_

Round 1
Reviewer 1 Report
Comments and Suggestions for Authors
First of all, I would like to congratulate the authors for the great work they have done and the excellent result.
The manuscript analyzes the latest advances in the study of osteoporosis proteomics. A very thorough analysis of recent publications in the area was performed, and a detailed list of markers that were used in the diagnosis and treatment of pathology was compiled.
The proposed for publication manuscript is original, citations have been carefully selected.
As a note - in the reviewer's opinion, a basic level is missing - the names of genes encoding proteins described as diagnostic and therapeutic markers. This would allow us to trace the entire chain and structure the presented data. However, this remark in no way reduces the quality of the work performed.
In general, this work undoubtedly structures the latest advances in the field of osteoporosis proteomics and can be recommended in present form.
Author Response
We are so glad to hear that the reviewer found our Manuscript of the latest advances in osteoporosis proteomics to be thorough and that the compilation of diagnostic and therapeutic markers was detailed.
In response to the reviewer's suggestion, we have made appropriate modifications to the manuscript.
We extend our gratitude to the reviewer for their diligent review and valuable feedback. We are confident that the revisions made will further improve the quality and readability of our manuscript.
Reviewer 2 Report
Comments and Suggestions for Authors
The manuscript "Proteomic Insights into Osteoporosis: Unraveling Diagnostic Markers and Therapeutic Targets for the Metabolic Bone Disease" authored by Wang and Xue et al. provides a comprehensive and well-structured analysis of the application of proteomic techniques in understanding osteoporosis. The review effectively summarizes the most recent advancements in proteomic studies related to osteoporosis, with a particular emphasis on the identification of potential prognosis biomarkers for osteoporosis and the discovery of novel therapeutic targets. Furthermore, the authors conclude the review by discussing the challenges and future scope in the proteomic field, highlighting the potential impact of proteomic research in transforming the landscape of osteoporosis diagnosis and treatment.
The review is well-documented and supported by recent literature references, making the content valuable for general readers and particularly beneficial for researchers involved in bone-related diseases. However, I would like to suggest some suggestions that could further strengthen the manuscript's quality. It would be beneficial for the authors to provide a brief discussion about the advancements in proteomics related to rare bone diseases, as this area is not extensively covered in the present manuscript. The inclusion of such content would contribute to a more comprehensive understanding of proteomic developments in osteoporosis and rare bone diseases. Overall, the manuscript effectively fulfills the objective of providing insights into proteomic advancements in osteoporosis.
Comments on the Quality of English LanguageMinor formatting and editing are required. Otherwise, the English language is fine and understandable.
Author Response
As the authors of the manuscript, we extend our sincere appreciation to the reviewer for their positive evaluation and insightful suggestions for improving our work.
In response to the reviewer's suggestion, we have made appropriate modifications to the manuscript to include a brief discussion about the advancements in proteomics related to rare bone diseases ("3.4.Advancements in Proteomics Related to Rare Bone Disease"). We believe that this addition enhances the comprehensiveness of our review and provides readers with a more holistic understanding of proteomic developments in both osteoporosis and rare bone diseases.
Reviewer 3 Report
Comments and Suggestions for Authors
The authors provide a review article which summarizes the current state of the art regarding the impact of proteomic approaches that have been performed in the context of skeletal disorders. The topic of the article is undoubtedly relevant, and the authors have accomplished to clarify the principal clinical relevance of proteomic studies for bone research. They also provide many examples obtained in several studies, some of them listed in Table 1. Therefore, this review article is surely informative and comprehensive. There are however some issues that could be improved, especially since it appears that many different observations have been reported, for which a diagnostic and therapeutic impact remains to be confirmed.
Specific comments:
1) After the introduction of the relevance regarding proteomics in bone research, section 2 essentially describes the generally applied technologies in proteomics, i.e. the subheading “Proteomic Technologies in Osteoporosis Research” is somehow misleading. It might be preferable to link this paragraph to the very informative Figure 1, which is only mentioned in the Conclusion.
2) In the remaining part of the main manuscript (section 3-5) the authors adequately present specific findings of potential clinical relevance in the text as well as a list of main findings in the context of osteoporosis biomarker identification displayed in Table 1. Although this way of referring to the literature is surely comprehensive, it appears that the majority of studies remained on a descriptive level, i.e. the clinical relevance of the putative biomarkers listed in Table 1 is still unknown. Moreover, since there is no real overlap in the biomarkers that have been identified in different studies, there should be at least some critical discussion about the reason, why (just to give one example) for instance CDH-13 was identified as a potential therapeutic factor in preventing osteoporosis only in one study, but not in other proteomic screening approaches. Although there is some discussion about the limitations of proteomic studies in section 6.1. this doesn´t necessarily explain why the majority of studies identify a completely different set of potentially relevant molecules compared to the others.
3) While there is no doubt about the principal importance of proteomic technologies in osteoporosis research, it would be informative, if possible, to provide at least one example, where the initial identification of a molecule by a proteomic screening approach led to follow-up investigations to further support its pathophysiological relevance. In other words, is there any example from bone research, where proteomic insights have already affected diagnosis and/or treatment of skeletal disorders? If not, it might be useful to provide such as an example from another clinically relevant disorder affecting other organs.
Comments on the Quality of English LanguageThe manuscript is overall well-written
Author Response
We thank the reviewer so much and sincerely appreciate the reviewer's thorough evaluation and constructive feedback on our work. We have made appropriate revisions to address the issues identified:
-
We have revised the section on "Proteomic Technologies in Osteoporosis Research" to better align it with the content and purpose of the manuscript. Additionally, we have linked this section to Figure 1 to provide readers with a visual representation of the technologies discussed.
-
We have added critical discussions about the clinical relevance and variability of biomarkers identified in different studies. Furthermore, we have addressed the issue of certain molecules are identified in some studies but not in others, providing possible explanations and limitations of proteomic studies.
-
The procollagen type-I N-terminal propeptide (PINP) stands as a prime example of a valuable biomarker in the clinical assessment of osteoporosis. As a precursor molecule of collagen type I, the primary structural protein in bone, PINP is released during the synthesis of type I collagen, offering insights into the activity of osteoblasts, the cells responsible for bone formation.
In clinical practice, the examination of PINP levels serves as an invaluable tool in the management of osteoporosis. It provides crucial information regarding bone turnover dynamics, treatment response, and fracture risk assessment. Reference #119 (Development of a highly sensitive, high-throughput, mass spectrometry-based assay for rat procollagen type-I N-terminal propeptide (PINP) to measure bone formation activity) exemplifies a study focused on PINP based on proteomic analysis, highlighting its significance in osteoporosis research.
Overall, we believe that these revisions have strengthened the manuscript by addressing the reviewer's comments and enhancing the clarity and depth of the discussion.